# Subgingival Microbiota of Mexicans with Type 2 Diabetes with Different Periodontal and Metabolic Conditions

**DOI:** 10.3390/ijerph16173184

**Published:** 2019-08-31

**Authors:** Adriana-Patricia Rodríguez-Hernández, María de Lourdes Márquez-Corona, América Patricia Pontigo-Loyola, Carlo Eduardo Medina-Solís, Laurie-Ann Ximenez-Fyvie

**Affiliations:** 1Laboratory of Molecular Genetics, School of Dentistry, National Autonomous University of Mexico (UNAM), Mexico City 04360, Mexico; 2The Academic Area of Dentistry in the Health Sciences Institute, the Autonomous University of the State of Hidalgo, Pachuca 42039, Mexico

**Keywords:** Type 2 diabetes mellitus, periodontitis, subgingival microbiota, “Checkerboard” DNA–DNA hybridization, obesity, glycemic poor control, dyslipidemia

## Abstract

Background: Type-2-Diabetes (T2D) and Periodontitis are major inflammatory diseases. However, not much is known about the specific subgingival microbiota in Mexicans with diabetes and metabolic dysbiosis. The aim of this study was to describe the subgingival microbiota of Mexicans with T2D and the different periodontal and metabolic conditions, through “Checkerboard” DNA–DNA hybridization. Methods: Subjects were divided into two groups—periodontal-health (PH) (PH_non-T2D; *n* = 59, PH_T2D; *n* = 14) and generalized-periodontitis (GP) (GP_non-T2D; *n* = 67, GP_T2D; *n* = 38). Obesity (BMI ≥ 30 kg/m^2^) and serum levels of glycated-hemoglobin (HbA1c), total-lipids, triglycerides, total-cholesterol, high-density-lipids, and low-density-lipids were measured for the T2D individuals. Subgingival microbial identification was processed for 40 species through DNA-probes. Results: Subjects with T2D harbored significantly higher mean total levels (PH: *p* < 0.001, and GP_NS), a lower proportion of “red” complex (GP: *p* < 0.01), a higher proportion of “yellow” (GP; *p* < 0.001), and “orange” (GP; *p* < 0.01) complex than the non-T2D. GP_T2D individuals exhibited a greater proportion of putative-species—*Campylobacter gracilis* and *S. constellatus* (*p* < 0.001), and *Parvimonas micra* and *Prevotella nigrescens* (*p* < 0.01), than GP_non-T2D. T2D individuals with HbA1c > 8% had presented significantly higher mean pocket-depth and higher levels of *G. morbillorum* (*p* < 0.05) and those with obesity or dyslipidemia harbored higher levels, prevalence, or proportion of *Streptococcus* sp., *Actinomyces* sp., and *Capnocytophaga* sp. Conclusions: T2D individuals harbored a particular microbial profile different to non-T2D microbiota. Metabolic control was related to dysbiosis of microbiota—HbA1c>8% related to periodontitis and obesity or dyslipidemia with the predominance of saccharolytic bacteria, irrespective of their periodontal condition.

## 1. Introduction

Worldwide, diabetes mellitus (DM) was found to have a prevalence of about 9% by 2014, and millions of deaths have resulted due to complications arising from this disease [1]. Mexico has the fifth highest rate of DM (12%) around the world with a Type 2 diabetes mellitus (T2D) incidence rate of 14.4% and 13.7% for males and females, respectively. An incidence rate of 28.1% of obesity (body mass index ≥30 kg/m^2^) in the global adult population has been recognized in upper middle-income countries [2,3]. A variety of chronic and metabolic diseases are related to DM, and some metabolic conditions including obesity, dyslipidemia, overweight, and obesity are crucially linked. Additionally, lifestyle factors like physical inactivity, unhealthy diet, tobacco smoking, and alcohol consumption are considered risk factors that increase prolonged states of hyperglycemia and lead to mortality from diabetes [2].

Evidence supports significant associations between periodontal inflammation, glycemic control, and dyslipidemia, and there is a growing evidence of an independent association between periodontitis and the incidence of diabetes [4,5]. Microbial subgingival composition of T2D individuals with periodontitis have been identified in past studies, which found the highest prevalence of periodontal pathogens like *Porphyromonas gingivalis* and *Tannerella forsythia* [6,7,8,9]. However, recent studies on The Human Oral Microbiome Database (HOMD) were found to be contradictory. These studies report a higher prevalence of putative periodontal pathogens like *Fusobacterium* sp., *Aggregatibacter* sp., and *Prevotella* sp. of T2D subjects with periodontitis [7,10]. In addition to the above, there is no evidence if dysbiosis of subgingival microbiota occurs in presence of metabolic conditions like poor glycemic control, obesity, or dyslipidemia of T2D individuals and it remains unclear whether specific subgingival microbial profiles can distinguish between diabetic and non-diabetic (non-T2D) individuals [4,11].

The aim of this study was to describe the subgingival microbiota of T2D subjects with periodontal-health (PH) and generalized-periodontitis (GP) and compare it with non-T2D individuals and similar periodontal condition, using the “Checkerboard” DNA–DNA hybridization technique, and to determine the microbial composition in different metabolic levels of glycemic control, lipid profile, and obesity of T2D subjects.

## 2. Materials and Methods

### 2.1. Subject Population

The study was approved by the Ethics Committee of the Health Sciences Institute at Autonomous University of the State of Hidalgo (Cinv/o/048/2016). Procedures were described to subjects prior to requesting their signature on informed-consent forms, acknowledging their willingness to participate. Subjects were randomly selected from patients at the Periodontology Department of UNAM and the Diabetes Prevention and Detection Clinic affiliated to the University of the State of Hidalgo (UAEH).

The population consisted of 178 subjects classified as periodontal-health (PH_non-T2D; *n* = 59, PH_T2D; *n* = 14) and generalized-periodontitis (GP_non-T2D; *n* = 67, GP_T2D; *n* = 38). Subjects had not received any periodontal therapy in the past, were non-current smokers, presented at least 18 teeth (excluding third molars), and were born in Mexico.

Through medical evaluations, diabetic subjects were diagnosed with T2D at least one year prior to their inclusion in the study (7.5 ± 0.8 years since diagnosis, mean ± standard error of the mean [SEM]) and exhibited levels of glycated hemoglobin (HbA1c) ≥ 6.5% on at least two separate tests. Classification of individuals was according to the most recent classification to periodontal conditions, considering their extent and distribution [12]. PH subjects had ≤4 sites with attachment levels (AL) of 4 mm, no sites with AL ≥ 5 mm or suppuration (SUP), mean full-mouth pocket-depth (PD) ≤ 3 mm, ≤40% of sites with bleeding on probing (BOP), and were of at least 18 (non-T2D) and 35 (T2D) years of age. GP subjects had ≥30% sites with AL ≥ 5 mm and were of at least 35 years of age. Exclusion criteria included current smokers, pregnancy, lactation, antibiotic-therapy in the 3 months prior to sampling, and any systemic condition (except T2D in the corresponding groups), such as HIV/AIDS or autoimmune diseases. Subjects receiving exogenous-insulin or presenting type 1 diabetes were also excluded. General and clinical characteristics of the subject population are summarized in Table 1.

### 2.2. Clinical Evaluation and Sample Collection

Clinical data and samples were collected per subject, on a single visit, by a calibrated clinician. General information obtained included medical history, age, gender, smoking status, weight, height, blood pressure, BMI (kg (weight)/m^2^ (height)), overweight (BMI: 25.0–29.9), obesity (BMI: 30.0–34.9, class I; 35.0–39.9, class II; ≥40.0, class III) and recent systemic antibiotic-therapy. Specific T2D information included time since diagnosis and pharmacological treatment for glycemic control. Periodontal parameters included previous periodontal treatment, number of missing teeth and clinical measurements taken at 6 sites per tooth from all teeth excluding third molars (168 sites per subject), as previously described [13]. Measurements assessed were PD (mm), AL (mm), plaque accumulation, (PLA, 0/1—undetected/detected), gingival erythema (GE, 0/1), BOP (0/1), and SUP (0/1). PD and AL were recorded twice by the same examiner, using a North Carolina periodontal probe. The average of the pair of measurements was used in the analyses.

After drying and isolation with cotton rolls, supragingival plaque was removed with curettes and subgingival plaque samples were obtained with individual sterile Gracey curettes from the mesio-buccal site, for up to 28 teeth (excluding third molars). Samples were placed in individual tubes containing 150 µL of TE buffer (10 mM Tris-HCl, 0.1 mM EDTA, pH 7.6), dispersed, and 100 µL of 0.5 M NaOH was added to each tube. A 5 mL sample of peripheral blood was collected from T2D subjects using a standard venipuncture technique and were placed in 16 × 100 mm glass vacutainer tubes containing sodium heparin (Becton, Dickinson, and Co., Franklin Lakes, NJ, USA). Tubes were inverted several times and processed to determine serum levels of HbA1c, total-lipids (TL), triglycerides (TG), total-cholesterol (TC), high-density-lipids (HDL), and low-density-lipids (LDL).

### 2.3. Microbial Assessment

Digoxigenin labeled whole-genomic DNA-probes were prepared using a random primer technique [14]. Samples were processed individually for the detection and enumeration of 40 microbial species through “Checkerboard” DNA–DNA hybridization [15], following the procedures previously described. The list of bacterial strains employed for the development of the DNA-probes is presented in Table 2. DNA was isolated and purified [16] from American Type Culture Collection (Rockville, MD) lyophilized stocks. Specificity and sensitivity of the DNA-probes were assessed, and the sensitivity of the assay was adjusted to approximately 10^4^ cells of each species.

### 2.4. Statistical Analyses

Statistical descriptive analysis of age, gender, height, weight, body weight, BMI, overweight, obesity, obesity I, II, III, serum levels of HbA1c, and clinical periodontal measurements—PD, AL, PLA, GE, BOP, SUP—are shown in Table 1; each expressed by their mean ± SEM. Clinical parameters differences were sought between the paired comparisons of the diabetic groups (separately for PH and GP), as well as between the periodontal groups, and separately for the non-T2D and T2D groups, assessed by Mann-Whitney U (MW) test.

Microbial data available were the absolute counts of each of the 40 test species from up to 28 subgingival plaque samples per subject (mean = 25.3 samples/subject, total = 4,205 samples). The analyses compared the composition of subgingival plaque between clinical groups, expressed as mean levels (DNA-probe count) ± SEM, prevalence (% sites colonized) ± SEM, and proportion (% total DNA-probe count) ± SEM of individual species. The proportion of microbial complexes was also determined by grouping the test species as similarly as possible to previous descriptions of microbial complexes, in subgingival plaque [17,18]. Exceptions are noted in Table 2. Each data type was computed for individual species and for microbial complexes in every sample, averaged within a subject, and then across subjects, in each group. Microbial differences were sought between paired comparisons between diabetic groups, separately for PH and GP, and between the periodontal groups (non-T2D and T2D), assessed by the MW test.

T2D individuals were separate between paired comparisons in obesity (BMI ≥ 30, *n* = 40) and non-obese T2D (BMI < 30, *n* = 12) as well as subjects with complete serum levels results distributed and compared between the HbA1c ≤ 8% (*n* = 31) versus HbA1c > 8% (*n* = 17); TL ≤ 800 mg/100 mL (*n* = 38) versus TL > 800 mg/100 mL (*n* = 10); TG < 150 mg/100 mL (*n* = 18) versus TG ≥ 150 mg/100 mL (*n* = 30); TC < 185 mg/100 mL (*n* = 25) versus TC ≥ 185 mg/100 mL (*n* = 23); HDL > 45 mg/10 (*n* = 9) versus HDL ≤ 45 mg/100 mL (*n* = 39); LDL < 100 mg/10 (*n* = 9) versus LDL ≥ 100 mg/100 mL (*n* = 39); and atherogenic index—LDL/HDL < 3 mg/10 (*n* = 15) versus LDL/HDL ≥ 3 mg/100 mL (*n* = 33). Clinical parameters and microbial data were sought between the clinical measurements groups in the different serum levels previously mentioned.

In all statistical tests, values of significance were adjusted for multiple comparisons, as previously described [19].

## 3. Results

The mean general information and clinical parameters (± SEM) of the 4 clinical groups are summarized in Table 1. Differences that were statistically significant between PH groups were age, AL (*p* < 0.001), PLA (*p* < 0.01), and BOP (*p* < 0.05) and between GP groups were age, PD, PLA, SUP (*p* < 0.001), and AL (*p* < 0.01) (non-T2D versusT2D). The main differences between periodontal groups (SP versus GP) showed that GP individuals harbored a higher mean of PD, AL, number sites with AL ≥ 5 mm, SUP (*p* < 0.001, non-T2D and T2D), age, BOP (*p* < 0.001, non-T2D, and *p* < 0.01, T2D), and number of missing teeth (*p* < 0.001, non-T2D). The general population presented a mean over 57.58% of overweight and obesity. Additionally, the GP_T2D group harbored the highest mean of the mentioned parameter than the rest of groups (71.05%).

From the total microbial samples (*N* = 4205) each of the 4 clinical groups represented a considerable number of samples for microbial comparisons and all of the test species were detected in the non-T2D (PH: *n* = 1400 and GP: *n* = 1551) and the T2D samples (PH: *n* = 353 and GP: *n* = 1551).

Figure 1 represents mean total levels of bacterial species (total DNA-probe count × 10^5^ ± SEM) in all subject groups. T2D subjects harbored higher mean total levels than non-T2D in both PH and GP groups. However, the difference was statistically significant only in PH (PH—non-T2D 133.16 ± 17.72 versus T2D 268.01 ± 40.63, *p* < 0.001, and GP—non-T2D 128.77 ± 11.96 versus GP_T2D 349.25 ± 23.26, NS). No significant differences were detected between the PH and GP subjects with either non-T2D or T2D systemic conditions (Appendix A).

Mean individual total levels (total DNA-probe count × 10^5^) of the 40 bacterial species showed in Figure 2, had presented higher levels in all species of T2D groups, with the exception *T. forsythia*, which showed higher mean levels for non-T2D groups (PH and GP). The difference was statistically significant in the T2D groups, which harbored higher mean individual levels than the non-T2D of species *Actinomyces georgiae, Actinomyces israelii, Actinomyces naeslundii* (GP non-T2D versus T2D, *p* < 0.001)*, Actinomyces viscosus* (PH non-T2D versus T2D, *p* < 0.05, and GP non-T2D versus T2D, *p* < 0.001), *Streptococcus anginosus* (PH non-T2D versus T2D, *p* < 0.05, and GP non-T2D versus T2D, *p* < 0.001), *Streptococcus gordonii* (PH non-T2D versus T2D, *p* < 0.05), *Streptococcus intermedius* (PH non-T2D versus T2D, *p* < 0.05)*, Streptococcus sanguinis* (PH non-T2D versus T2D, *p* < 0.01, and GP non-T2D versus T2D, *p* < 0.001), *Veillonella parvula* (GP non-T2D versus T2D, *p* < 0.001), *Capnocytophaga ochracea* (PH non-T2D versus T2D, *p* < 0.05, and GP non-T2D versus T2D, *p* < 0.001), *Capnocytophaga sputigena, Campylobacter gracilis* (PH non-T2D versus T2D, *p* < 0.01)*, Campylobacter showae* (PH non-T2D versus T2D, *p* < 0.05), *Campylobacter rectus, Fusobacterium nucleatum, Fusobacterium periodonticum, Parvimonas micra*, and *Aggregatibacter actinomycetemcomitans* (GP non-T2D versus T2D, *p* < 0.001), *Streptococcus constellatus* (PH non-T2D versus T2D, *p* < 0.01, and GP non-T2D versus T2D, *p* < 0.001)*, Corynebacterium matruchotii* (GP non-T2D versus T2D, *p* < 0.01), *Eubacterium saburreum* (PH non-T2D versus T2D, *p* < 0.01, and GP non-T2D versus T2D, *p* < 0.001)*, Neisseria mucosa* (PH non-T2D versus T2D, *p* < 0.05)*,* and *Selenomonas artemidis* (PH non-T2D versus T2D, *p* < 0.05, and GP non-T2D versus T2D, *p* < 0.001) (Figure 2). Significant differences were detected between PH and GP with either non-T2D or T2D, only with higher individual levels of *P. gingivalis* in the non-T2D—(PH versus GP, *p* < 0.001), and T2D (PH versus GP, *p* < 0.05) (Figure 2, Appendix A).

The mean prevalence (% sites colonized) of the 40 individual test-species showed a higher percent in the T2D groups, irrespective of their periodontal condition. PH_T2D subjects exhibited a significantly higher mean prevalence than the PH_non-T2D subjects of the species, namely *A. naeslundii* (PH—non-T2D 69.1% ± 3.3 versus T2D 92.3% ± 2.2, *p* < 0.05)*, C. rectus* (PH—non-T2D 32.4% ± 3.7 versus T2D 62.9% ± 6.6, *p* < 0.05)*,* and *S. artemidis* (PH—non-T2D 35.5% ± 4.2 versus T2D 68.4% ± 6.2, *p* < 0.05). The GP_T2D group exhibited a significantly higher mean prevalence than the GP_non-T2D of 27 out of the 40 evaluated species (Appendix A). The statistically significant differences between GP and PH subject in non-T2D was due to a higher prevalence of *A. naeslundii* (non-T2D—PH 69.1% ± 3.3 versus GP 46.2% ± 4.3, *p* < 0.05), *P. gingivalis* (non-T2D—PH 41.5% ± 4.3 versus GP 63.9% ± 3.5, *p* < 0.01), and *T. forsythia* (non-T2D—PH 38.6% ± 4.0 versus GP 55.9% ± 3.2, *p* < 0.05). No statistically significant differences were observed between PH and GP in the T2D groups.

The mean proportions (% total DNA-probe count) of microbial complexes in each clinical group are summarized in Figure 3. PH and GP groups exhibited higher proportions of the red-complex species in the non-T2D individuals than in the T2D ones (GP non-T2D—19.1% versus GP T2D—9.3%, *p* < 0.01), and in the “yellow” (GP non-T2D—6.8% versus T2D—14.1%, *p* < 0.001), “green” (GP non-T2D—5.9% versus T2D—6.1%, NS), and “orange” (GP non-T2D—17.9% versus T2D—22.8%, *p* < 0.01) complex species with a higher proportion in the T2D groups than in the non-T2D (Figure 3).

Mean individual proportions of the 40 evaluated species are summarized in Figure 4.

Mean individual proportions of the 40 evaluated species are summarized in Figure 4. Subjects with GP_T2D exhibited significantly higher proportions than the non-T2D of species *A. georgiae, S. anginosus*, *S. intermedius, Streptococcus mitis, Streptococcus oralis* (*p* < 0.001)*, S. sanguinis* (*p* < 0.01)*, V. parvula* (*p* < 0.05)*, C. ochracea, C. gracilis, S. constellatus* (*p* < 0.001)*, P. micra, Prevotella nigrescens* (*p* < 0.01)*, Selenomonas noxia* (*p* < 0.01), and *S. artemidis* (*p* < 0.001) (Figure 4, Appendix A). The statistically significant differences between the GP and PH subject in non-T2D were due to a higher proportion of “red” complex (*p* < 0.01), and *P. gingivalis* (*p* < 0.001) species, and a lower proportion of “*Actinomyces*” complex (*p* < 0.05), *A. naeslundii* (*p* < 0.01), “yellow” complex, and *S. mitis* (*p* < 0.05) species. No statistically significant differences were observed between PH and GP in the T2D individuals (Figure 3 and Figure 4) and between PH non-T2D and T2D (Appendix A).

Clinical parameters and statistically significant microbial data between clinical measurements groups in obesity and different serum levels of HbA1c subjects are shown in Table 3. No statistically significant differences observed between the T2D individuals with BMI < 30 and BMI ≥ 30 with other periodontal clinical measurements or serum levels. However, all periodontal measurements presented a higher mean percent in obesity and in individuals with HbA1c > 8% but only the mean pocket-depth presented significantly higher mean percent in poor glycemic control (*p* < 0.01) (Table 3).

Individual mean levels (Lev), prevalence (Prev), or proportion (Pr) ± SEM of the 40 species evaluated for T2D subjects with obesity (BMI ≥ 30), exhibited significantly increased levels of bacterial species *S. intermedius* (Lev_*p* < 0.01, and Pr_*p* < 0.05), *T. denticola* (Lev_*p* < 0.05), and *N. mucosa* (Lev_*p* < 0.05), as compared to individuals with a BMI < 30. Microbial data also showed a significantly higher mean levels of *Gemella morbillorum* in individuals with HbA1c > 8% (Table 3).

Microbial data differences were sought between paired comparisons with different lipid profile levels; shown in Table 4.

Subjects harbored statistically significant differences in TL poor control, with a higher mean of *A. israelii* (Lev_*p* < 0.05), *A. naeslundii* (Prev_*p* < 0.01)*, Capnocytophaga gingivalis* (Prev_*p* < 0.05 and Pr_*p* < 0.01)*, E. saburreum* (Prev_*p* < 0.05)*, E. sulci* (Prev_*p* < 0.01)*, S. gordonii* (Prev_*p* < 0.05); in TC poor control with higher mean percent of *A. naeslundii* (Pr_*p* < 0.05)*, C. gingivalis* (Prev_*p* < 0.01), *C. matruchotii* (Pr_*p* < 0.01), *E. sulci* (Prev_*p* < 0.05)*, F. periodonticum* (Prev_*p* < 0.05), *S. gordonii* (Prev_*p* < 0.05), and *T. forsythia* (Pr_*p* < 0.05)*;* in LDL poor control with higher mean of *A. actinomycetemcomitans* (Lev_*p* < 0.05); in LDL/HDL risk with a higher mean percent of *A. viscosus* (Pr_*p* < 0.05) (Table 4). No statistically significant differences were observed in the rest of the species evaluated.

## 4. Discussion

In the general and clinical periodontal evaluation of the present study, subjects with T2D exhibited significantly more sites with PLA, BOP, and SUP than non-T2D (PH and GP). According to previous studies, BOP and SUP are indicators of periodontal disease risk and severity in T2D [4,20]. However, real periodontal damage is evaluated with periodontal attachment loss as part of the periodontal diagnosis [21]; clinical AL of T2D individuals in the present evaluation exhibited significant less mean values than those without T2D. The rest of the paired differences, with exception of age, showed a similar distribution between the non-T2D and T2D groups. In the comparative analyses between subjects with PH and GP, we determined that individuals harbored differences according to the actual periodontal classification (considering extent and distribution) [12]. The discrepancies in the “n” of each clinical group studied, did not represent a problem for clinical or microbial comparison, since sample collection included full mouth microbial evaluation by standardized curette sampling [22], including analysis per sample. Despite the fact that T2D individuals represents a higher risk of periodontal diseases [4], we included PH T2D group, considering the relevance to periodontal comparisons with PH non-T2D and GP T2D individuals, strongly recommended by the literature [23].

Our results indicated that while some differences could be detected consistently between the T2D and the non-T2D subjects, significant differences in subgingival microbial profiles were observed between both diabetic and periodontal groups. Subgingival plaque from T2D and non-T2D harbored significantly greater levels in the T2D groups (PH and GP), irrespective of the periodontal condition, this was considered the most important difference reported in a study of the T2D individuals with periodontitis [24], but the main evidence to the present investigation was contradictory, with a particular microbiota of the T2D individuals. While some studies had reported similar subgingival microbiota in periodontitis of T2D and non-T2D individuals [25,26], contradictory recent reports using molecular techniques, have suggested that specific microorganisms, including the “non-periodontal pathogenic” species *S. sanguinis*, *S. oralis*, *S. intermedius,* and *Actinomyces* sp. harbor higher levels or prevalence in subgingival plaque of T2D individuals [7,27]. The results of the present study are in accordance with those last reports, with a predominance of levels, prevalence, and proportion of *Actinomyces* sp. and *Streptococcus* sp. in T2D groups (in PH and GP), compared to the non-T2D one. Therefore, our findings suggested a distinct subgingival microbial profile for T2D individuals, with few differences between PH and GP.

In the GP_T2D subjects, the highest proportion the “orange” and “yellow” complex species (not considering the complex “other”) and a lesser proportion of the “red” complex compared to GP non-T2D individuals and the specific “non periodontal pathogenic” related species of the “purple” (*V. parvula*) were observed. The “green” complex (*C. ochracea*), periodontal putative “orange” (*C. gracilis, S. constellatus, P. micra*, and *P. nigrescens*,), and “other” complex microbial species (*S. noxia and S. artemidis*) presented higher mean levels and proportion in the GP_T2D subjects, in accordance with previously described scientific literature, as studied by sequencing the identification of the HOMD. An association of some microorganisms with a higher occurrence of specific species like *Veillonella* sp. [7]. *Campylobacter* sp. [28], and *Capnocytophaga* sp. [7] was also observed in the subgingival plaque of T2D subjects with periodontitis, in addition to the high prevalence of *P. micra* and *Prevotella* sp. in the subgingival plaque of pre-diabetic subjects with insulin resistance [29]. These main results indicated a dysbiosis of the subgingival microflora of T2D individuals, which is certainly related to their systemic condition. T2D individuals of the present investigation in both PH and GP groups, presented mean levels of HbA1c over 8.17%, with the highest percent of overweight and obesity.

The subgingival microbiota described in previous studies [6,7,30,31] is currently recognized as the most frequently associated with chronic and generalized periodontitis. It has been characterized by higher proportions of “red” complex species and low proportions of *Actinomyces* sp. compared to periodontal-health in individuals with non-metabolic risk factors; these reports were in accord to subgingival microbiota of non-T2D individuals of the present study. On the other hand, similar findings of a lower prevalence of “red” complex species *Porphyromonas* sp. and *Tannerella* sp. have been reported in subgingival microbiota of T2D individuals with periodontitis [18,32]. Accordingly, our results indicated that the differences between non-T2D and T2D individuals were modest in the proportion of the pathogenic species *P. gingivalis*, *T. forsythia*, and *T. denticola.* T2D individuals of the present study harbored less significant proportion of “red” complex in GP compared to non-T2D and differences between PH and GP were only in *P. gingivalis*, which presented higher mean levels and prevalence in GP_T2D. Recent studies have reported on the subgingival microbiota of individuals with T2D and other metabolic conditions. These microbial results had reported a “more simple” or “non-complex” microbiota in the number and kind of species, with a higher variability of putative periodontal species as compared to non-metabolic compromised individuals [10,29,33]. These findings were in accord with the results of the present study, which allows us to infer the main role of the putative periodontal species in the development of GP in T2D individuals with a very particular microbial profile.

To date, there are no studies that have reported on the changes in the microbiota of T2D individuals, which have considered metabolic characteristics such as obesity, poor glycemic control, and dyslipidemia, and have analyzed them as separate risk factors [28,34,35]. Additionally, it remains unclear if the levels of HbA1c and dyslipidemia are related to changes in the subgingival microbiota of subjects with T2D [11,28,36]. Nevertheless, a strong association between periodontitis and poor control of HbA1c (>8%) and lipid profile was reported [11,37,38,39]. According to the present study, the findings with all clinical periodontal measurements had higher mean values in T2D individuals with obesity and HbA1c > 8% showed a strong relation between periodontitis and T2D individuals with such metabolic disorders. We also determined that T2D individuals with higher obesity levels and proportion of *S. intermedius*, and with HbA1c > 8% exhibited significantly higher levels of *G. morbillorum* but not the other 40 evaluated species. These results were contradictory to other reports that found that subjects with obesity show higher proportions of some pathogenic species [40,41] and another study that found that subjects with poor glycemic control, with mean value of HbA1c > 10%, showed higher abundances of *Streptococcus* sp. [42]. Our contradictory findings might be a result of the differences in glycemic control of T2D subjects.

Our results indicated that microbial dysbiosis of T2D were related by factors beyond just obesity or glycemic poor control, when considering the subjects who presented with dyslipidemia disorders. However, we could infer from the present study that subjects with TL, TC poor control, and an atherogenic index harbored an additional risk factor that influenced microbial changes of T2D, with a higher proportion of saccharolytic species like *Streptococcus* sp., *Actinomyces* sp., and *Capnocytophaga* sp., in accordance with the subgingival microbiota of previous reports [35,42]. Despite the fact that putative or pathogenic periodontal species *F. periodonticum*, *T. forsythia*, and *A. actinomycetemcomitans* presented higher levels, prevalence, or proportion in T2D individuals with TC and LDL poor control, and *T. denticola* in obesity (as observed in this study), it was not clear when those pathogenic species did not comprise the main microbiota in GP. According to the scientific literature, metabolic and nutritional status of diabetic individuals are linked to subgingival microbial dysbiosis due to dysregulated immune-inflammatory responses, and the subgingival environment represent an enriched source of carbohydrates [10,33,42]. Therefore, the need for a longitudinal study to evaluate the transition of subgingival microbiota from PH to GP in T2D is evident; however, from the results of the present study we could be speculated that T2D might share certain immunological traits that could lead to comparable changes in the subgingival microbial composition, due to obesity and poor metabolic control of glycated hemoglobin and lipid profile. Although our results add to the state of clinical and epidemiological knowledge, there are some limitations in our research—primarily, that cross-sectional studies cannot establish causal relationships between dependent and independent variables due to its temporal ambiguity.

## 5. Conclusions

The results of the present study suggest differences in the subgingival microbiota in accordance with diabetic status observed. We identified a particular subgingival microbial profile associated with subjects with GP and T2D, with higher levels, prevalence, or proportion of putative periodontal species of the “orange” complex, which probably play the main role in pathogenicity and are related to periodontal disease severity. Additionally, species of the “red” complex play a main pathogenic role in GP of non-T2D individuals, contradictory to the subgingival microbial profile of T2D groups, which exhibit a higher proportion of non-pathogenic species in both periodontal conditions (PH and GP). The distinctive subgingival microbial dysbiosis of T2D subjects was associated with metabolic control. Additionally, HbA1c > 8% related to periodontitis, obesity, or dyslipidemia harbored a particular niche in the colonizing-saccharolytic-bacteria, irrespective of the periodontal condition.

## Figures and Tables

**Figure 1 ijerph-16-03184-f001:**
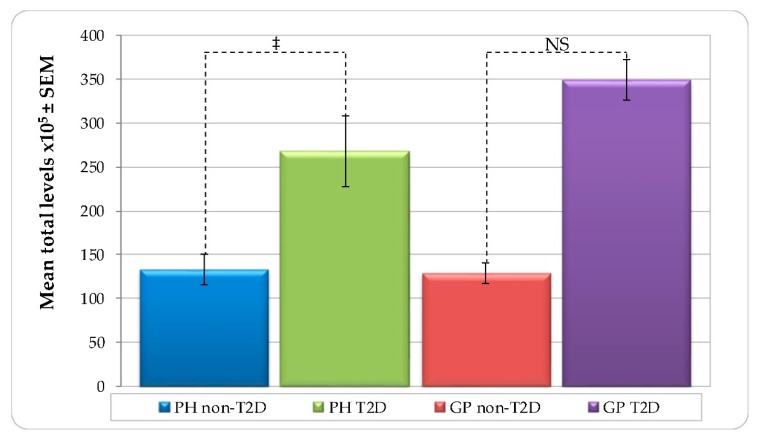
Mean total levels per group (total DNA-probe count × 10^5^ ± SEM) of subgingival plaque samples from 178 non-T2D and T2D subjects (PH_non-T2D, *n* = 59; PH_T2D, *n* = 1; GP non-T2D, *n* = 67; and GP T2D, *n* = 38). Paired differences were determined by the Mann–Whitney test. The difference between the PH and GP non-T2D versus T2D were observed in each group. ‡ *p* < 0.001. NS—not significant.

**Figure 2 ijerph-16-03184-f002:**
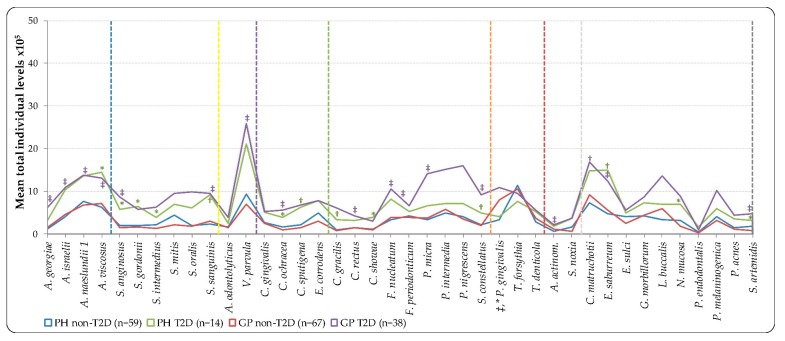
Mean total individual levels (total DNA-probe count × 10^5^) of 40 individual bacterial species subgingival plaque samples from 178 non-T2D and T2D subjects (PH_non-T2D, *n* = 59; PH_T2D, *n* = 14; and GP non-T2D, *n* = 67; GP T2D, *n* = 38). Total levels were computed in every sample, averaged within a subject, and then across subjects in each group. Taxa were grouped by color lines according to the descriptions of microbial complexes in the subgingival plaque [17,18], exceptions are noted in Table II). Paired differences were determined by the Mann–Whitney test, after adjusting for 40 comparisons, as previously described [19]. The difference between PH and GP non-T2D versus T2D (greater significances found in group lines); PH versus GP non-T2D and T2D (greater significances found in labels) were * *p* < 0.05, † *p* < 0.01, ‡ *p* < 0.001. NS—Not significant; *A. actinom*—*Aggregatibacter actinomycetemcomitans*.

**Figure 3 ijerph-16-03184-f003:**
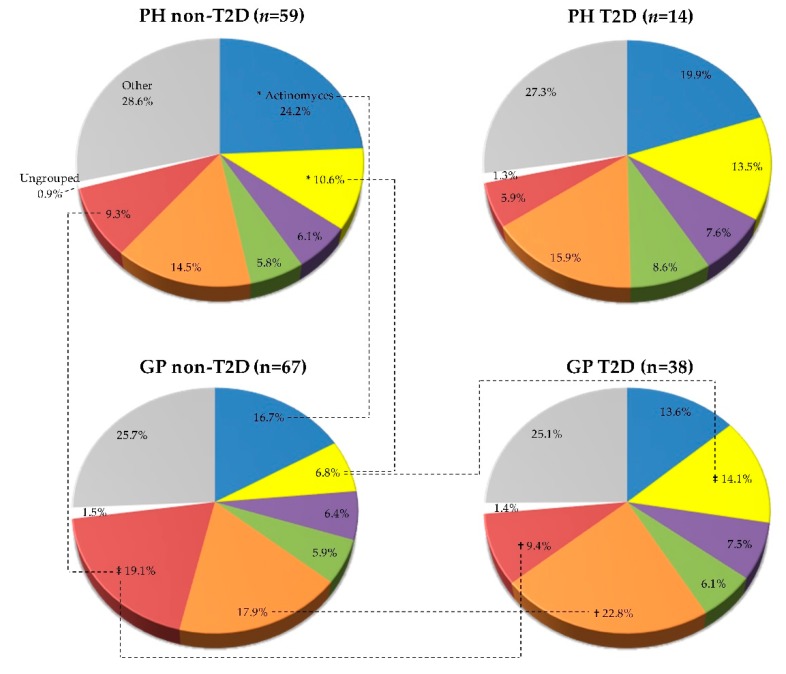
Mean proportion (% total DNA-probe count) of microbial complexes subgingival plaque samples from 178 non-T2D and T2D subjects (PH_non-T2D, *n* = 59; PH_T2D, *n* = 14; and GP non-T2D, *n* = 67; GP T2D, *n* = 38). Taxa were grouped according to the descriptions of microbial complexes in the subgingival plaque [17,18]. Paired differences were determined by the Mann–Whitney test, after adjusting for 8 comparisons, as previously described [19]. The difference between the PH and GP non-T2D versus T2D: * *p* < 0.05, † *p* < 0.01, ‡ *p* < 0.001. The differences in the T2D groups (PH vs. GP) were not statistically significant.

**Figure 4 ijerph-16-03184-f004:**
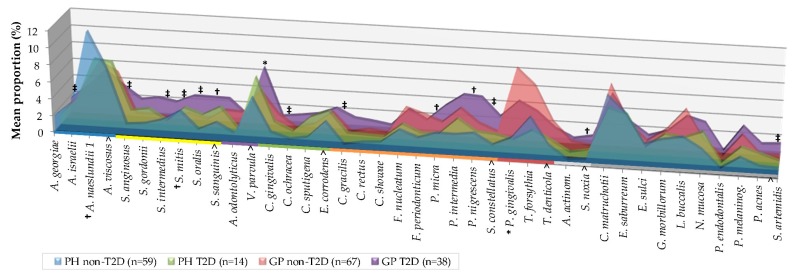
Mean proportion (% total DNA-probe count) of 40 individual bacterial species subgingival plaque samples from 178 non-T2D and T2D subjects (PH_non-T2D, n = 59, PH_T2D, *n* = 14; GP non-T2D, *n* = 67; and GP T2D, *n* = 38). The proportion of each species was computed in every sample, averaged within a subject, and then across subjects in each group. Taxa were grouped by colored lines, according to the descriptions of the microbial complexes in the subgingival plaque [17,18] (exceptions are noted in Table II) and are presented alphabetically within each complex. Paired differences were determined by the Mann–Whitney test after adjusting for 40 comparisons, as previously described [19]. The difference between GP non-T2D versus T2D (greater significances found in the GP-T2D group); PH versus GP non-T2D (greater significances found in labels) were * *p* < 0.05, † *p* < 0.01, ‡ *p* < 0.001. The differences in PH and T2D groups were not statistically significant. NS—Not significant; *A. actinom*—*Aggregatibacter actinomycetemcomitans*.

**Table 1 ijerph-16-03184-t001:** General and clinical characteristics of the subject population (*N* = 178).

Variable	PH Non-T2D (*n* = 59)			PH T2D (*n* = 14)				GP Non-T2D (*n* = 67)			GP T2D (*n* = 38)			
	Media		SEM	Media		SEM	MW	Media		SEM	Media		SEM	MW
Age (years) ‡ †	31.39	±	1.20	46.00	±	2.10	‡	49.10	±	1.24	59.47	±	1.77	‡
Gender (% female)	59.32	±		57.14	±			53.73	±		68.42	±		
Height (cm)	160.86	±	1.97	159.57	±	2.69		160.50	±	1.46	151.08	±	1.50	
Weight (kg)	68.53	±	2.12	69.93	±	3.11		70.36	±	2.56	64.11	±	2.33	
Body fat (%) *	35.22	±	2.09	31.11	±	1.94		27.11	±	1.41	30.69	±	1.20	
Mean body mass index	26.58	±	0.76	27.45	±	1.09		26.91	±	0.74	27.98	±	0.81	
Overweight and obesity (%)	57.58	±		64.29	±			59.09	±		71.05	±		
Obesity I, II, and III (%)	21.21	±		21.43				18.18			23.68			
Glycated hemoglobin (%)		-		8.17	±	0.52					8.31	±	0.34	
Number of missing teeth ‡	1.29	±	0.21	2.00	±	0.59		3.37	±	0.29	3.82	±	0.41	
Mean pocket-depth (mm) ‡ €	2.16	±	0.04	2.32	±	0.09		3.90	±	0.12	3.19	±	0.13	‡
Mean attachment level (AL, mm) ‡ €	1.89	±	0.04	1.40	±	0.08	‡	4.32	±	0.15	3.45	±	0.17	†
Mean number sites with AL ≥ 5 mm ‡ €	0	±	0	0	±	0		50.52	±	3.64	35.97	±	4.37	
% sites with:														
Plaque accumulation	45.88		4.92	86.28		4.11	†	51.71		4.30	92.86		1.53	‡
Gingival erythema	19.87	±	3.46	22.81	±	8.04		25.00	±	3.70	34.81	±	3.49	
Bleeding on probing ‡ †	16.35	±	2.54	30.34	±	3.75	*	45.95	±	2.76	58.13	±	3.79	
Suppuration ‡ €	0	±	0	0	±	0		4.91	±	0.91	18.34	±	2.51	‡

Paired differences were determined by the Mann–Whitney U (MW column) test: * *p* < 0.05, † *p* < 0.01, ‡ *p* < 0.001 PH and GP non-T2D versus T2D, * *p* < 0.05, ‡ *p* < 0.001 PH non-T2D versus GP non-T2D, and † *p* < 0.01, € *p* < 0.001 PH T2D versus GP T2D (significances in labels). PH—periodontal-health; GP—generalized-periodontitis; non-T2D—non-diabetic individuals; T2D—Type-2-Diabetes; SEM—Standard Error of Mean.

**Table 2 ijerph-16-03184-t002:** Reference strains employed for the development of the DNA-probes.

Species	ATCC	Complex	Species	ATCC	Complex
*Actinomyces georgiae*	49285	*Actinomyces*	*Neisseria mucosa*	19696	Other
*Actinomyces israelii*	12102	*Actinomyces*	*Parvimonas micra*	33270	Orange
*Actinomyces naeslundii*	12104	*Actinomyces*	*Porphyromonas endodontalis*	35406	Other
*Actinomyces odontolyticus*	17929	Purple	*Porphyromonas gingivalis*	33277	Red
*Actinomyces viscosus*	43146	*Actinomyces*	*Prevotella intermedia*	25611	Orange
*Aggregatibacter actinomycetemcomitans*	*****	Ungrouped	*Prevotella melaninogenica*	25845	Other
*Campylobacter gracilis*	33236	Orange	*Prevotella nigrescens*	33563	Orange
*Campylobacter rectus*	33238	Orange	*Propionibacterium acnes*	6919	Other
*Campylobacter showae*	51146	Orange	*Selenomonas artemidis*	43528	Other
*Capnocytophaga gingivalis*	33624	Green	*Selenomonas noxia*	43541	Ungrouped
*Capnocytophaga ochracea*	27872	Green	*Streptococcus anginosus*	33397	Yellow
*Capnocytophaga sputigena*	33612	Green	*Streptococcus constellatus*	27823	Orange
*Corynebacterium matruchotii*	14266	Other	*Streptococcus gordonii*	10558	Yellow
*Eikenella corrodens*	23834	Green	*Streptococcus intermedius*	27335	Yellow
*Eubacterium saburreum*	33271	Other	*Streptococcus mitis*	49456	Yellow
*Eubacterium sulci*	35585	Other	*Streptococcus oralis*	35037	Yellow
*Fusobacterium nucleatum*	******	Orange	*Streptococcus sanguinis*	10556	Yellow
*Fusobacterium periodonticum*	33693	Orange	*Tannerella forsythia*	43037	Red
*Gemella morbillorum*	27824	Other	*Treponema denticola*	35405	Red
*Leptotrichia buccalis*	14201	Other	*Veillonella parvula*	10790	Purple

***** Serotypes a (43717) and b (43718) were combined to generate a single DNA-probe. ****** Subspecies *nucleatum* (25586), *polymorphum* (10953), and *vincentii* (49256) were combined to generate a single DNA-probe. (ATCC—American Type Culture Collection, Rockville, MD. Complex—species were grouped acording to the descriptions of microbial complexes in subgingival plaque [17,18], with the following exceptions—*C. matruchotii*, *E.*
*saburreum*, *E. sulci*, *G.*
*morbillorum*, *L.*
*buccalis*, *N. mucosa*, *P.*
*endodontalis*, *P.*
*melaninogenica*, *P. acnes*, and *S.*
*artemidis* were grouped as “Other”).

**Table 3 ijerph-16-03184-t003:** Clinical characteristics and statistically significant microbial data of T2D subjects.

Parameter	Obesity (−) (*n* = 40)			Obesity (+) (*n* = 12)			MW	HbA1c ≤ 8% (*n* = 31)			HbA1c >8% (*n* = 17)			MW
	Media		SEM	Media		SEM		Media		SEM	Media		SEM	
Mean pocket-depth (mm)	2.9	±	0.1	3.1	±	0.2		2.7	±	0.1	3.3	±	0.2	†
Mean attachment level (AL, mm)	2.9	±	0.2	2.9	±	0.3		2.6	±	0.2	3.2	±	0.3	
Mean number sites with AL ≥ 5 mm	25.6	±	4.3	28.7	±	9.2		19.5	±	3.7	36.5	±	7.9	
Plaque accumulation	90.4	±	2.0	93.3	±	1.9		89.0	±	2.5	94.2	±	1.6	
Gingival erythema	31.4	±	3.9	32.3	±	7.0		27.3	±	3.9	32.1	±	5.1	
Bleeding on probing	50.4	±	3.9	51.5	±	7.4		44.7	±	3.7	58.1	±	6.5	
Suppuration	13.5	±	2.4	14.6	±	4.8		10.8	±	2.5	18.8	±	4.0	
Total lipids (mg/L)	663.7	±	24.6	677.3	±	63.8		643.2	±	24.6	710.0	±	48.6	
Triglycerides (mg/100 mL)	203.8	±	16.2	303.6	±	131.1		190.2	±	16.4	293.2	±	84.8	
Total cholesterol (mg/100 mL)	196.1	±	9.0	196.9	±	16.0		191.5	±	8.7	205.1	±	15.3	
High Density Lipids (HDL)	38.9	±	1.5	40.1	±	2.7		40.3	±	1.6	37.0	±	2.3	
Low Density Lipids (LDL)	142.4	±	7.8	145.9	±	15.2		139.1	±	7.7	150.7	±	13.7	
Atherogenic index (LDL/HDL)	3.8	±	0.2	3.9	±	0.7		3.5	±	0.2	4.3	±	0.5	
Species														
*S. intermedius*_levels	4.4	±	0.4	9.9	±	1.8	†	5.0	±	0.6	7.0	±	1.4	
*S. intermedius*_proportion	1.5	±	0.2	2.2	±	0.4	*	1.7	±	0.3	1.7	±	0.3	
*T. denticola*_levels	5.0	±	0.8	7.4	±	1.4	*	4.3	±	0.7	6.3	±	1.1	
*N. mucosa*_levels	6.4	±	0.8	14.3	±	4.2	*	8.0	±	1.2	9.8	±	3.0	
*G. morbillorum*_levels	7.7	±	0.9	10.3	±	2.4		7.2	±	1.1	10.8	±	1.7	*

Paired differences were determined by the Mann–Whitney U (MW) test; * *p* < 0.05, † *p* < 0.01; SEM: Standard Error of the Mean.

**Table 4 ijerph-16-03184-t004:** Statistically significant microbial data of T2D subjects with different lipid profile levels.

Species	TL ≤ 800 mg/100 mL (*n* = 38)			TL > 800 mg/100 mL (*n* = 10)			MW	TC < 185 mg/100 mL (*n* = 25)			TC ≥ 185 mg/100 mL (*n* = 23)			MW	LDL < 100 mg/100 mL (*n* = 9)			LDL ≥ 100 mg/100 mL (*n* = 39)			MW	LDL/HDL < 3 mg/100 mL (*n* = 15)			LDL/HDL ≥ 3 mg/100 mL (*n* = 33)			MW
	Media		SEM	Media		SEM		Media		SEM	Media		SEM		Media		SEM	Media		SEM		Media		SEM	Media		SEM	
*A. israelii*_levels	10.17	±	2.36	15.60	±	4.15	*	8.08	±	1.49	14.81	±	3.92		7.57	±	2.59	12.17	±	2.46		8.68	±	2.11	12.50	±	2.85	
*A. naeslundii*_prevalence	79.47	±	3.27	95.82	±	2.04	†	79.66	±	4.35	86.37	±	3.33		65.86	±	10.00	86.80	±	2.19		77.40	±	6.89	85.36	±	2.56	
*A. naeslundii*_proportion	5.03	±	0.80	9.84	±	3.31		3.79	±	0.45	8.46	±	1.83	*	3.81	±	0.78	6.54	±	1.16		3.66	±	0.52	7.11	±	1.34	
*A. viscosus*_proportion	4.22	±	0.86	5.80	±	1.43		4.15	±	1.26	4.98	±	0.75		3.54	±	0.71	4.79	±	0.90		2.74	±	0.44	5.38	±	1.03	*
*A. actinomycetem.*_levels	2.20	±	0.33	1.81	±	0.66		2.33	±	0.50	1.89	±	0.30		1.11	±	0.25	2.35	±	0.35	*	1.79	±	0.32	2.27	±	0.41	
*C. rectus*_proportion	1.12	±	0.15	0.66	±	0.20		1.31	±	0.21	0.73	±	0.13		1.22	±	0.42	0.98	±	0.13		1.20	±	0.32	0.95	±	0.13	
*C. gingivalis*_prevalence	47.88	±	4.87	74.36	±	10.63	*	39.88	±	4.48	67.73	±	7.32	†	45.56	±	9.76	55.39	±	5.32		42.98	±	6.87	58.45	±	5.93	
*C. gingivalis*_proportion	1.01	±	0.22	2.05	±	0.50	†	0.77	±	0.09	1.72	±	0.40		0.77	±	0.19	1.34	±	0.25		0.74	±	0.12	1.46	±	0.30	
*C. matruchotii*_proportion	5.76	±	0.86	7.41	±	1.05		4.40	±	0.48	7.95	±	1.32	†	5.50	±	0.86	6.24	±	0.86		4.66	±	0.59	6.75	±	1.00	
*E. saburreum*_prevalence	63.95	±	4.60	80.37	±	8.79	*	62.46	±	5.53	72.70	±	6.18		53.41	±	11.86	70.59	±	4.24		63.84	±	8.16	68.97	±	4.83	
*E. sulci*_prevalence	50.87	±	4.69	79.74	±	7.27	†	48.50	±	5.17	66.00	±	6.69	*	51.93	±	10.10	58.03	±	4.84		54.17	±	6.86	58.12	±	5.53	
*F. periodonticum*_prevalence	64.25	±	5.30	79.69	±	7.30		60.79	±	5.87	75.72	±	6.70	*	59.03	±	11.37	69.69	±	4.89		65.90	±	7.59	68.43	±	5.67	
*S. gordonii*_prevalence	65.36	±	5.05	85.47	±	8.53	*	63.30	±	5.74	76.08	±	6.74	*	62.91	±	10.11	71.56	±	5.05		73.85	±	5.79	68.02	±	6.01	
*T. forsythia*_proportion	2.88	±	0.43	3.54	±	1.06		2.20	±	0.46	3.90	±	0.64	*	2.91	±	1.16	3.04	±	0.43		2.48	±	0.71	3.26	±	0.49	

TL—total lipids; TC—total cholesterol; LDL—low density lipids; LDL/HDL—atherogenic index; *A. actinomycetem—Aggregatibacter actinomycetemcomitans*; Paired differences were determined by the Mann–Whitney U (MW) test; * *p* < 0.05, † *p* < 0.01.

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
