# Peer review of "Subgingival Microbiota of Mexicans with Type 2 Diabetes with Different Periodontal and Metabolic Conditions"

_ijerph, 2019, doi:10.3390/ijerph16173184_

Round 1

Reviewer 1 Report

The article is complete and well written. Only minor typing errors are present.

Author Response

The article is complete and well written. Only minor typing errors are present.

R: Thanks. We have implemented changes to address this recommendation

Reviewer 2 Report

The authors have developed interesting work related to periodontal disease and diabetes. Yet major revisions are suggested to improve the quality of the manuscript in regards to additional data stratification and text corrections. 

Terminology: Periodontal diseases and diabetes are major inflammatory and metabolic diseases and not degenerative. Please correct this throughout the text. 

Please correct the technique description throughout the text in regards to DNA-DNA checkboard. It needs to be consistent in one format.

Introduction: While social economic status (SES) is important, the authors have not included that subanalysis in the study. Please include a supplementary analysis with stratification of SES status -similar to table 1.

Results: It is unnecessary to provide individual mean levels in the text. A table summarizing the results will provide clarity to the text. Results are meant to provide a description of the findings, while discussion a comparison to the literature. Please revise.

table one: Please include HbA1C levels.

Please revise the following: page 11, lines 269-271. Why are you not showing the not statistically significant differences?  Please provide a stratified data analysis related to BMI and obesity. 

Discussion: Provide an explanation for the discrepancies in the n of the study. Specially PH T2D had much lower numbers. Were the dropouts? Is there any specific reasoning in recruitment? 

Please discuss the HBA1C results.

Conclusion: Include a sentence related to the fact this a cross-sectional association study and no causality can be found in this study.

he improvements suggested will provide better quality for the readership of the Journal.

Author Response

The authors have developed interesting work related to periodontal disease and diabetes. Yet major revisions are suggested to improve the quality of the manuscript in regards to additional data stratification and text corrections.

Terminology: Periodontal diseases and diabetes are major inflammatory and metabolic diseases and not degenerative. Please correct this throughout the text.

R: Correction lines 15-16. There is evidence that Periodontitis is not degenerative, but it is destructive, we correct the misconception.

Please correct the technique description throughout the text in regards to DNA-DNA checkboard. It needs to be consistent in one format.

R: Correction lines 19, 35-36, 63, 116, Checkerboard” DNA-DNA hybridization; Additional correction: line 23, 119 DNA-probes.

Introduction: While social economic status (SES) is important, the authors have not included that subanalysis in the study. Please include a supplementary analysis with stratification of SES status -similar to table 1.

R: It is interesting this suggestion for the present study, unfortunately, we did not ask the subjects individual information related to SES. Government of Santiago de Anaya Hidalgo had reported low income, with 72.8% of poverty in their population by 2010, however, we don´t have the same information of sample collections of Mexico City citizens, living in 16 different municipalities. Global information in the Introduction “Upper middle income” in Mexico, are related to “Obesity” status because of changes in their correct nutrition. We believe this is a good point for further research to ask in the sampling population.

Results: It is unnecessary to provide individual mean levels in the text. A table summarizing the results will provide clarity to the text.

R: Additional Table S1: Mean total individual levels (total DNA-probe count x 105) of 40 individual bacterial species subgingival plaque samples (N= 178). Correction: lines 172, 197, 211, 242, 246-7, and 379-383 Supplementary Materials. We removed the levels in lines: 182-194.

Results are meant to provide a description of the findings, while discussion a comparison to the literature. Please revise.

R: We added Table 3 and Table 4 to express detailed results of microbial data.

Discussion section provide corrections related to the suggestion. Lines: 281-367.

table one: Please include HbA1C levels.

R: Serum levels of HbA1c included in Table 1 (information proper of diabetic subjects). Corrections: lines 86-88, 128.

Please revise the following: page 11, lines 269-271. Why are you not showing the not statistically significant differences?  Please provide a stratified data analysis related to BMI and obesity.

R: We decide to add “data not showed previously”. We add table 3 in order to clarify clinical data with and without statistically significant differences. Corrections lines 144, 149, 248-256, and 257-271.

Discussion: Provide an explanation for the discrepancies in the n of the study. Specially PH T2D had much lower numbers. Were the dropouts? Is there any specific reasoning in recruitment?

R: The present study included only 14 PH non-T2D individuals because of their uncommon PH due to their systemic condition, however we add some information related to number of sample collection. Additional results lines: 134, 163-166 and in discussion section lines: 281-287.

In order to clarify discrepancies in “n” of serum levels we add corrections line: 144.

Please discuss the HBA1C results.

R: Corrections lines: 312-315 and 337-343.

Additional corrections: Abstract section lines 23-31.

Conclusion: Include a sentence related to the fact this a cross-sectional association study and no causality can be found in this study.

R: Correction “Although our results add to the state of clinical and epidemiological knowledge, there are some limitations in our research – primarily, that cross-sectional studies cannot establish causal relationships between dependent and independent variables because of temporal ambiguity.”.

Additional corrections: Abstract section lines 31-34.

The improvements suggested will provide better quality for the readership of the Journal.

R: We are grateful for the Reviewer for valuable comments.

Reviewer 3 Report

This manuscript mainly described the subgingival microbiota of Mexicans with type 2 diabetes with different periodontal and metabolic conditions. The test species were detected in the 4 clinical groups: periodontal health_non-T2D, periodontal health_T2D, generalized-periodontitis_non-T2D, generalized-periodontitis_T2D. The microbial assessment was evaluated by checkerboard DNA-DNA hybridization.

The strength of this study is that the general and clinical periodontal characteristics of the subjects were well-collected and presented. Supragingival plaque was removed before subgingival plaque were collected. The way of tables and figures are also presented clear. There are no major issues with the presentation or conclusion. However, there are a couple of minor issues needed to be concern:

1. The number of periodontal health with T2D is lower than the other groups, the detailed procedure of recruitment or the reasons of drop out would be better to be described.

2. Table 2 for DNA-probes is mislabeled as Table 1 and need to be corrected.

3. The way of expression in the Results section needs to be improved. From reader perspective, summarized results presented in a table would be more clear than describing all mean levels through text.

4. It would be interesting to add more discussion to interpret the association between metabolic status of T2D and subgingival microbial dysbiosis.

Author Response

Reviewer 3

Comments and Suggestions for Authors

This manuscript mainly described the subgingival microbiota of Mexicans with type 2 diabetes with different periodontal and metabolic conditions. The test species were detected in the 4 clinical groups: periodontal health_non-T2D, periodontal health_T2D, generalized-periodontitis_non-T2D, generalized-periodontitis_T2D. The microbial assessment was evaluated by checkerboard DNA-DNA hybridization.

The strength of this study is that the general and clinical periodontal characteristics of the subjects were well-collected and presented. Supragingival plaque was removed before subgingival plaque were collected. The way of tables and figures are also presented clear. There are no major issues with the presentation or conclusion. However, there are a couple of minor issues needed to be concern:

The number of periodontal health with T2D is lower than the other groups, the detailed procedure of recruitment or the reasons of drop out would be better to be described.

R: The present study included only 14 PH non-T2D individuals because of their uncommon PH due to their systemic condition. Additional information of microbial samples per group had showed in results, lines: 134, 163-166 and in discussion section lines: 281-287 to clarify that point as you suggest.

In order to clarify discrepancies in “n” of serum levels we add corrections line: 144.

Table 2 for DNA-probes is mislabeled as Table 1 and need to be corrected.

R: Correction line 121 Table 2.

The way of expression in the Results section needs to be improved. From reader perspective, summarized results presented in a table would be more clear than describing all mean levels through text.

R: We add Table 3 line 253 and table 4 line 261 in order to express microbial results clearly.

Corrections in results section, lines: 251-260 and 264-271.

Additional corrections: Abstract section lines 23-31.

It would be interesting to add more discussion to interpret the association between metabolic status of T2D and subgingival microbial dysbiosis.

R: We add corrections in discussion section, lines: 312-315, 342-350 and 358-367.

Corrections for discussion, line 294: (T2D individuals); line 300 (additional text); Lines 303 Change order: (non-considering the complex “other”); line 326 (mean levels); 328 (had reported); line 330 (those findings in accord); line 353: (change connector).

Round 2

Reviewer 2 Report

Authors have completed revisions successfully.